# DISCOVERING MINIMAL REINFORCEMENT LEARNING ENVIRONMENTS

## ABSTRACT

Human agents often acquire skills under conditions that are significantly different from the context in which the skill is needed. For example, students prepare for an exam not by taking it, but by studying books or supplementary material. Can artificial agents benefit from training outside of their evaluation environment as well? In this project, we develop a novel meta-optimization framework to discover neural network-based synthetic environments. We find that training contextual bandits suffices to train Reinforcement Learning agents that generalize well to their evaluation environment, eliminating the need to meta-learn a transition function. We show that the synthetic contextual bandits train Reinforcement Learning agents in a fraction of time steps and wall clock time, and generalize across hyperparameter settings and algorithms. Using our method in combination with a curriculum on the performance evaluation horizon, we are able to achieve competitive results on a number of challenging continuous control problems. Our approach opens a multitude of new research directions: Contextual bandits are easy to interpret, yielding insights into the tasks that are encoded by the evaluation environment. Additionally, we demonstrate that synthetic environments can be used in downstream meta-learning setups, derive a new policy from the differentiable reward function, and show that the synthetic environments generalize to entirely different optimization settings.

## 1 INTRODUCTION

Reinforcement Learning (RL) agents are commonly trained and evaluated in precisely the same environment. It is well known that this approach has several significant disadvantages: RL agents are brittle with respect to minor changes in the environment dynamics, hyperparameter choices, or even the concrete implementation of an algorithm (Henderson et al., 2018; Engstrom et al., 2019; Cobbe et al., 2020; Agarwal et al., 2021). Most recent research in RL has focused on improving RL algorithms in order to alleviate these challenges. But what about the Reinforcement Learning environment or the underlying Markov Decision Process (MDP) itself? Unlike RL agents, professional athletes train under vastly different conditions than their final competition settings. For example, long-distance runners do not repeatedly run the target distance, but train shorter interval runs, progressively increase their pace, and occasionally mix in long runs. Moreover, the development of sensory circuits in the brain is initially guided by "artificial stimuli" that are internally generated, before sensory stimuli from the environment become available (Katz & Shatz, 1996). Hence, the optimal environment dynamics for training may be drastically different from the final evaluation setting.

How can we apply these insights to training RL agents? Here, we leverage the recently proposed framework of synthetic environments (Ferreira et al., 2022) and show that complex tasks with complex transition dynamics and long time horizons can be greatly simplified by training agents on synthetic contextual bandit (SCB) tasks, referring to MDPs without state transition dynamics. This simplifies the approach of Ferreira et al. (2022), who learn a full state-transition function and omit learning the initial state distribution. To this end, we parameterize the distribution of initial states and the reward function of these synthetic environments by small neural networks and meta-learn their weights using evolutionary optimization. Training standard RL algorithms on these SCBs produces agents that generalize to the complex original task, which we refer to as the *evaluation environment* in the following.

The SCBs train agents in a fraction of time steps compared to training on the evaluation environment and provide a fast hardware-accelerated synthetic simulator (see Fig. 1, bottom). The individual environment components are all differentiable and we demonstrate their interpretability. Interestingly, we find that the synthetic reward function has learned which state dimensions are relevant to the optimal policy and varying irrelevant parts of the state leaves the learned reward invariant. The differentiable reward function encodes information about the reward-to-go in the evaluation environment, and can therefore be used to construct an "induced" policy. Furthermore, the costly meta-optimization process can be amortized in rapid downstream meta-learning applications and even generalizes to evolutionary optimization of agent policies. Our contributions are:

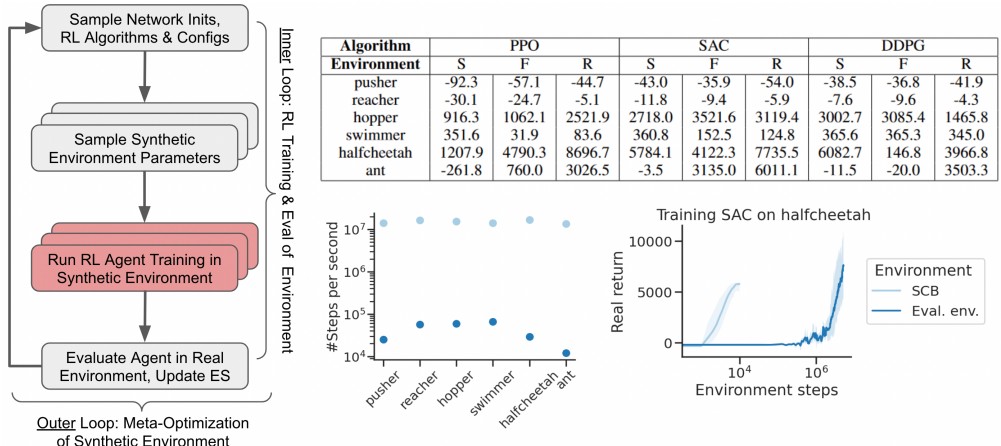

Figure 1: Discovering synthetic RL environments via meta-evolution. **Left**. Conceptual overview. We meta-evolve the parameters of a neural network-based environment using a task distribution over agent initializations, training algorithms, and hyperparameters. **Right, top**. The resulting synthetic environments are capable of training agents to high performance when evaluated on continuous control environments of Brax (Freeman et al., 2021). We report the return interquartile mean (IQM) over 50 runs. S and F: sampled & fixed inner loop tasks, respectively. R: tuned agent on the evaluation environment. **Right, bottom**. The neural networks underlying the SCB leverage hardware acceleration and are more compute-efficient (wall clock time) than the real dynamics, shown are steps per second when running 10000 steps on 1000 environments (10 million steps total) with random actions on and A100. The SCBs train agents in a fraction of the environment steps needed for training in the evaluation environment.

1. We introduce a meta-optimization framework for synthetic environment discovery leveraging contextual bandits with a learned initial state distribution and a curriculum on the evaluation length of the agents trained in the synthetic environment (Section 3).

2. We show that meta-training over a large range of inner loop tasks leads to synthetic environments that generalize across hyperparameters and other RL algorithms (Section 4).

3. The resulting CBs are interpretable (Section 5) and provide a direct way to probe the importance of individual state dimensions.

4. They can also be used for a plethora of downstream applications including the rapid meta-learning of policy optimization objective functions, policy derivation from the reward function, and even evolutionary optimization of agents (Section 6).

5. We release two open-source libraries accessible to the wider community:[1]

   - `synthetic-gymnax`: A repository of synthetic environments characterized by neural networks with pre-trained weight checkpoints.
   - `purerl`: A set of hardware-accelerated RL algorithms (SAC, PPO, DQN, DDPG, TD5) that run entirely on GPU/TPU which enables fast meta-optimization evaluation.

---

[1]The code and corresponding synthetic checkpoints will be released upon publication under https://github.com/<anonymous>/purerl and https://github.com/<anonymous>/synthetic-gym. Along this submission, we provide a single checkpoint & training in a notebook.

## 2 BACKGROUND & RELATED WORK

**Reinforcement Learning Formalism**. RL is interested in leveraging sampled agent experiences to solve an MDP (Puterman, 1990), i.e. to extract an optimal policy that maximizes the expected discounted cumulative return, $\mathbb{E}_\pi[\sum_{t=0}^T \gamma^t r_t]$. An MDP is defined as the tuple $\langle \mathcal{I}, \mathcal{S}, \mathcal{A}, \mathcal{T}, \mathcal{R}, d \rangle$. At the beginning of each episode an initial state $s_0 \sim \mathcal{I} \in \mathcal{S}$ is sampled. Afterwards, at each timestep $t$, an agent samples an action from its policy $a_t \sim \pi(\cdot|\mathrm{s}_t)$ (where $a_t \in \mathcal{A}$ and given a state $s_t \in \mathcal{S}$). The environment then issues a reward $\mathcal{R}(\mathrm{s}_t, \mathrm{a}_t)$ and updates the next state $\mathrm{s}_{t+1}$ according to the transition function $\mathrm{s}_{t+1} \sim \mathcal{T}(\cdot|\mathrm{s}_t, \mathrm{a}_t)$. An episode termination is indicated by a boolean $d(t, s, a)$ which in turn leads to the reset used for the next episode rollout. Throughout meta-training and evaluation we focus on a set of commonly used value- and policy-gradient based algorithms including DQN (Mnih et al., 2013), SAC (Haarnoja et al., 2018), PPO (Schulman et al., 2017), DDPG (Lillicrap et al., 2015), and TD3 (Fujimoto et al., 2018).

**Curricula for Reinforcement Learning**. Substantial amounts of effort have been put into designing curricula for RL agents. These include prioritization techniques (Schaul et al., 2015; Jiang et al., 2021), gradually increasing goal distances (Florensa et al., 2017), or learned sequencing methods (Narvekar & Stone, 2018). In this work, instead of manually designing a curriculum, we discover initial state distributions and reward functions maximizing the performance in the evaluation environment.

**Training Reinforcement Learning Agents with Synthetic Data**. Various methods for training machine learning models from synthetically generated data have been proposed. For example, this includes dataset distillation for supervised training (Wang et al., 2018) or synthetic experience replay for RL (Lu et al., 2023b). Applications for training with synthetic data include data augmentation and cheap data generation, which is especially important when requiring large amounts of data, such as in RL. Most closely related to our work is the approach outlined by Ferreira et al. (2022) which learns the reward- and state transition function while using the reset distribution of the original environment. They highlight that their approach struggles to generalize across broad ranges of hyperparameters and fails to scale to continuous control environments. Here, we demonstrate that it is possible to transform large MDPs into SCBs via meta-optimization for the first time.

**Meta-Optimization & Evolutionary Optimization**. Meta-optimization is commonly conducted using one of two approaches: Meta-gradient calculation with respect to a meta-objective or evolutionary black-box optimization of a fitness score. The calculation of higher-order gradients may fail for long unroll lengths and can result in myopic meta-solutions (Metz et al., 2021). Therefore, we leverage Evolution Strategies (ES) that adapt a parameterized distribution (e.g. multivariate normal) to iteratively find well-performing solutions. More formally, we use a search distribution $\mathcal{N}(\mu, \Sigma)$ with mean $\mu \in \mathbb{R}^{|\theta|}$ and a diagonal covariance matrix $\Sigma_{ij} = \sigma_i \delta_{ij}$, to sample candidate synthetic environments. After sampling a population of candidates, the fitness of each population member $f(x)$ is estimated using Monte Carlo evaluations. We use an aggregated fitness score summarizing the performance of the synthetic environments by evaluating a trained agent in the real environment. The scores are used to update the search distribution such that the expected fitness under the search distribution $\int_x f(x)\mathcal{N}(\mu, \Sigma)$ is maximized, according to SNES (Schaul et al., 2011).

**Discovering Algorithm Components via Evolutionary Meta-Learning**. Recently, the general combination of evolutionary optimization and neural network-based algorithm families has been used to discover various powerful algorithms. This includes the meta-discovery of gradient-based (Metz et al., 2022) and gradient-free (Lange et al., 2022; 2023) optimization algorithms, policy optimization objective functions (Lu et al., 2022), or reward functions (Faust et al., 2019). Furthermore, these synthetic artifacts can often be reverse-engineered to generate human-interpretable components. Here, we use the same paradigm to transform real environment simulators into SCBs.

**Hardware Accelerated Reinforcement Learning Environments**. Commonly, RL environments have been bound to CPUs and constrained by limited parallelism. Recently, there has been a paradigm change with RL simulators being accelerated by accelerator parallelism. These efforts include Brax (Freeman et al., 2021), Gymnax (Lange, 2022b), Jumanji (Bonnet et al., 2023), Pgx (Koyamada et al., 2023), or NVIDIA Isaac Gym (Makoviychuk et al., 2021). Still, most of them require the translation of the original step transition logic into hardware-specific coding frameworks (e.g. JAX (Bradbury et al., 2018)). Here, we provide a means to automatically yield hardware-accelerated neural-network-based environment proxies for training RL agents that generalize to potentially non-accelerated environments.

---

**Algorithm 1: Training Synthetic Environments with ES**

**Require:** Evaluation environment $E_t$
**Require:** Generations $T$, Rollouts $R$, ES $\mathcal{S}(\mu, \Sigma)$
**Require:** RL algorithms w. hyperparameters distrib.
**Require:** Evaluation episode length schedule $s$
  Initialize $\mu \sim$ network initialization, $\Sigma = \mathrm{diag}(\sigma)$
  **for** gen $= 1, \ldots, T$ **do**
    Sample population of environments $P \sim \mathcal{S}(\mu, \Sigma)$
    **for** Synthetic environment $E_s$ in $P$ **do**
      Select set of RL algorithms $A$
      Calculate evaluation length $l = s(\mathrm{gen})$
      **for** Algorithm algo in $A$ **do**
        **for** $r = 1, \ldots, R$ **do**
          Sample hyperparameter configuration
          Train agent $a$ in $E_s$ using algo /conf
          Get $f_{\mathrm{algo},r}$ as return of $a$ in $E_t(l, a)$.
        **end for**
      **end for**
      Fitness of $E_s$: $\frac{1}{|A|R} \sum_{\mathrm{algo} \in A} \sum_{r=1}^{R} f_{\mathrm{algo},r}$
    **end for**
    Update $\mu, \Sigma$ according to ES using fitness scores
  **end for**
  **return** synthetic environment with parameters $\mu$

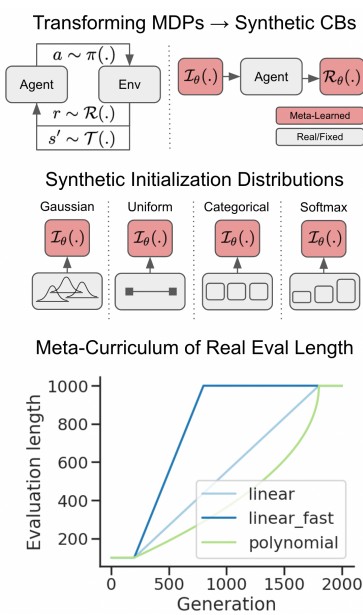

Transforming MDPs → Synthetic CBs

Synthetic Initialization Distributions

Meta-Curriculum of Real Eval Length

## 3 METHODS

**Synthetic Environment Setup**. RL environments are commonly modeled as Markov decision processes, consisting of a set of states $S$, a set of actions $A$, a distribution for the initial state $\mathcal{I}$, the reward function $\mathcal{R}(s, a)$, and the state transition function $\mathcal{T}(s'|s, a)$. We parameterize $\mathcal{I}_\theta$ and $\mathcal{R}_\theta(s, a)$ using a small neural network for each. To sample initial states, we calculate $s_0 = \mathcal{I}_\theta(z)$, where $z$ is a latent vector sampled from $z \sim P_z \in \mathbb{R}^n$. The choice of $P_z$ and $n$ are hyperparameters, which we set to $P_z = \mathcal{N}(0, I_n)$ and $n$ to be the dimensionality of the state space. The set of synthetic states is then given by the range of $\mathcal{I}_\theta$, while the set of synthetic actions is the same as the set of actions in the evaluation environment.

We omit parameterizing $\mathcal{T}(s'|s, a)$, such that synthetic environments become synthetic contextual bandits. This is conceptually different from Ferreira et al. (2022), who fix the initial distribution to be that of the evaluation environment, and learn the transition function instead. Training contextual bandits has several advantages: For example, it stabilizes the meta-training process since the recurrent forward pass of synthetic states through a neural network can lead to exploding values. Additionally, it significantly reduces the number of parameters from $\mathcal{O}(\dim(S)^2)$ to $\mathcal{O}(\dim(S))$, which eases the meta-training process.

Our choice of using CBs is justified by the fact that the optimal policy of any MDP can be found by policy optimization on a separate CB. Such a CB can be constructed by setting $r_{\mathrm{CB}}(s, a) = Q^*_{\mathrm{MDP}}(s, a)$ and $\mathcal{I}_{\mathrm{CB}} = U[S_{\mathrm{MDP}}]$. By maximizing the reward in the CB, a policy automatically maximizes the value function of the MDP in every state, and is therefore optimal when transferred. However, other choices of $r_{\mathrm{CB}}$ and $\mathcal{I}_{\mathrm{CB}}$ are possible to achieve optimal performance in practice. Its not necessary to correctly estimate the value of every state in the MDP, since some states might never be reached by an expert policy. Additionally, most policy optimization algorithms choose actions as $a = \arg\max_a Q(s, a)$, meaning that in order to perform well on the evaluation environment, the relative scale of rewards in the CB does not have to match the value estimates in the MDP. Discovering CBs therefore leaves several degrees of freedom, as the SCB can select states which are most relevant in learning evaluation task, and might scale rewards to quickly imprint a specific behavior. We empirically confirm the advantages of using SCBs in Appendix A.1, and present a comprehensive comparison between the meta-learned synthetic reward and the learned value function of an expert policy in Appendix A.2.

A list of hyperparameters for the synthetic environment can be found in Appendix B.1.

**Discovery via Meta-Evolution**. The parameters $\theta$ of the synthetic environment are meta-optimized using the separable natural evolution strategy (SNES, Schaul et al., 2011), implemented by `evosax` (Lange, 2022a). At each iteration of the meta-optimization algorithm (outer loop), we sample a population of synthetic environments according to the search distribution. We evaluate the fitness of each population member by training an agent in the synthetic environment (inner loop) and then calculating its return on multiple initializations of the evaluation environment. Subsequently, the fitness scores are used to update the search distribution according to SNES, such that the expected fitness under the search distribution is maximized.

In order to achieve generalization across algorithms and hyperparameters, we train multiple RL algorithms using a wide range of randomly sampled hyperparameter combinations in each meta-generation. We do so by vectorizing a training algorithm and then initializing with a vector of sampled hyperparameters. Thus, we are limited to parameters that can be vectorized over, i.e. whose values don't affect the memory layout or structure of compiled code. For a list of sampled hyperparameters see Appendix B.2.

**Meta-Evolution Fitness Evaluation Curriculum**. Many of the continuous control problems in Brax (Freeman et al., 2021), such as hopper or ant, require learning balance and locomotion. When calculating the fitness of synthetic environments using episodes of the full 1000 environment steps, they quickly converge to a local optimum of balancing the body while not moving forward. To address this issue, we use a curriculum on the length of the fitness evaluation rollout: We begin meta-training using short episodes in the real environment to evaluate fitness, and gradually increase their length. This ensures that the focus shifts towards locomotion early in meta-optimzation since the gain from balancing is limited. The overall meta-evolution process for synthetic environment discovery is outlined in Algorithm 1. In the following sections, we will probe and validate the following scientific questions:

1. Can we transform environments with multi-step MDPs into single-step SCBs with flexible reward and state initialization functions? What are the contributions of the meta-evolution design including the curriculum design and latent distribution for the initial state (Section 4)?

2. What are the properties of the resulting neural network-based SCBs? Can they be interpreted and potentially even provide insights into the underlying real environment dynamics (Section 5)?

3. How can we amortize the computationally expensive meta-discovery process? Is it possible to apply the synthetic environments to downstream applications with potential computational advantages and speed-ups (Section 6)?

## 4 RESULTS OF META-TRAINING

Fig. 2 shows the performance of synthetic environments that were meta-trained with multiple inner loop RL algorithms and sampled hyperparameter configurations. We were able to train SCBs for the challenging continuous control environments in the Brax suite, significantly extending the scope of results in Ferreira et al. (2022). The first row visualizes the meta-learning curves, where we indicate the fitness of the population mean. We noticed that for Halfcheetah, the inclusion of PPO in the set of RL algorithms made training unstable, likely because the range of sampled learning rates for PPO is too large for stable gradient-based optimization of the policy network. On the Swimmer environment, meta-training with sampled inner loop hyperparameters improves performance. This is likely because there are several distinct modes of behavior, and sampling hyperparameters introduces additional noise, such that new modes of behavior might be found more easily.

The second row shows the learning curves of RL agents when training in the SCB and evaluating in the evaluation environment. Notably, the agents achieve competitive performance on the Brax suite within 10000 time steps, whereas training in the evaluation environments typically takes several million time steps, and requires extensive hyperparameter tuning. The performance can be improved further by fixing the inner loop algorithm.

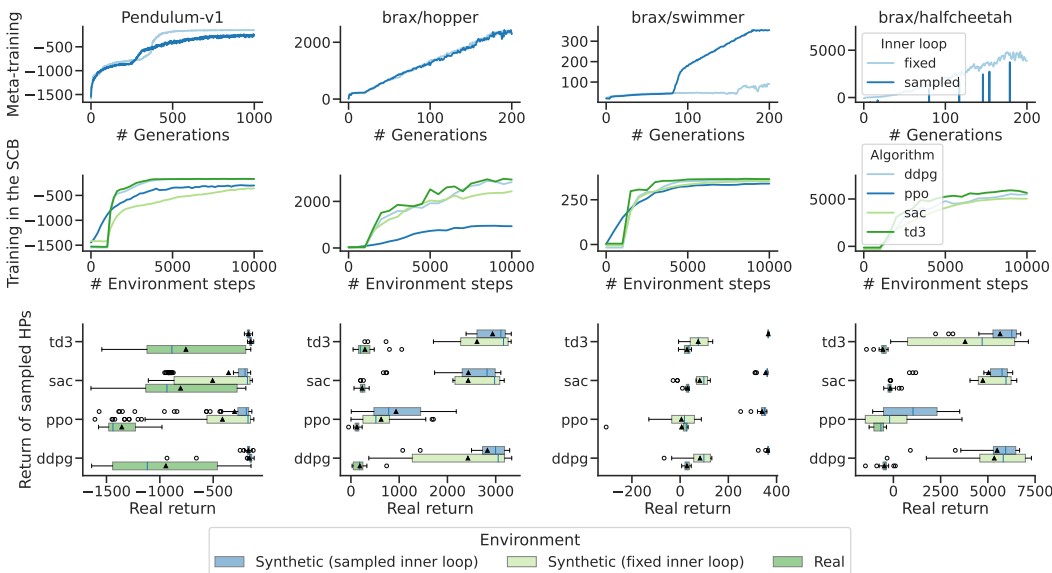

Figure 2: Meta-evolution, SCB evaluation, and agent hyperparameter robustness. **Top**. Our proposed meta-evolution setup enables the discovery of SCBs for challenging continous control environments for the first time. **Middle**. The discovered SCBs generalize across various common RL algorithms and train in few step transitions. **Bottom**. The SCBs are much more robust across hyperparameter settings than their real analogues, especially when sampling hyperparameters during meta-training. The evaluation results are aggregated as the IQM over 20 independent runs.

The third row shows the return distribution of agents with fixed/sampled hyperparameters on SCBs with fixed/sampled hyperparameters in the inner loop, as well as the evaluation environment. While SCBs generalize well, the vast majority of agents trained in the evaluation environments perform poorly, as they are usually very brittle with respect to their parameters. Achieving a good performance on challenging RL environments often requires additional hacks, such as observation- and reward normalization, extensions of the replay buffer (Schaul et al., 2015; Andrychowicz et al., 2017), generalized state-dependent exploration (Raffin et al., 2021), and others. These requirements are completely eliminated when training in the SCB.

Fig. 3 shows several ablations of our method. In the first row, we visualize four different meta-training settings, ingredients indicated by the presence of the letter

**T** for a parameterized transition function

**I** for a parameterized initial state distribution

**C** for the application of an evaluation episode length curriculum

The T setup acts as our main baseline, for which we closely mimic the setup of Ferreira et al. (2022) within our framework. This is necessary because we need to leverage our highly parallelizable implementation of RL algorithms to run experiments on Brax. For better comparison with different ablations, we increase the population size (16 to 64-256) and number of evaluation environments (10 to 64) to be consistent with our other ablations. Both changes are generally favorable to the performance (for details see Table 3).

The plain T setup is consistently beaten by our extensions. On MountainCar-v0, it is not able to discover an environment in which the agent reaches the goal, achieving a mean return of -200 on all evaluation seeds of all meta-training runs. It is well known that even state-of-the-art RL algorithms such as PPO struggle with solving MountainCar, due to the extremely sparse reward of reaching the flag, which is very improbable to achieve through random exploration.

Introducing a parameterized initial state distribution in TI circumvents this problem, as the environment can learn a distribution of relevant observations directly, without having to reach them via repeated application of the transition function. Omitting the transition function increases the performance on almost all classic control environments (see Appendix A.1).

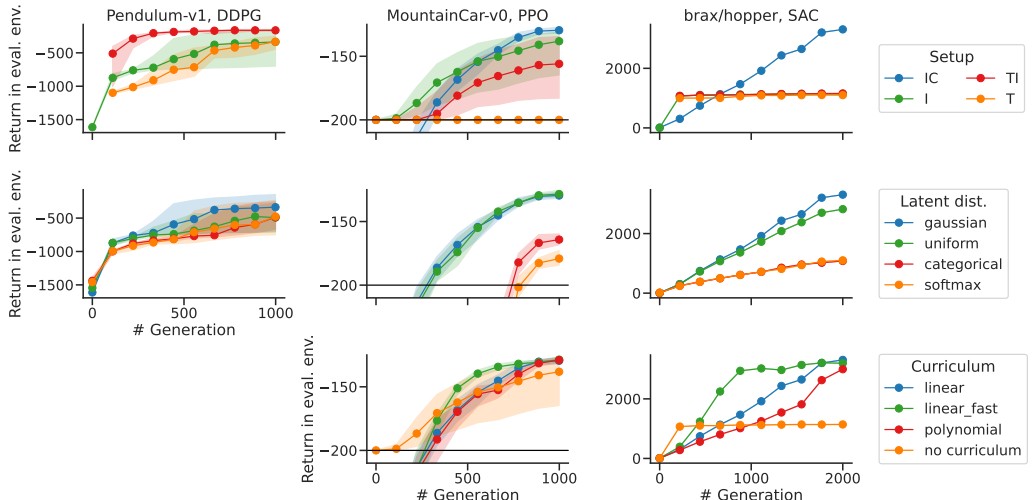

Figure 3: Ablation study evaluating meta-evolution ingredients on a specific environment-algorithm combination. **Top**. We compare the impact of parameterizing the initial state distribution (I), transition function (T), and the evaluation length curriculum (C). All three contributions lead to robust and scalable meta-discovery. **Middle**. Continuous latent distributions for the initial state distribution perform better than categorical ones. **Bottom**. The meta-training setup is robust the exact choice of evaluation episode length curriculum. The figure shows IQMs and 95% confidence intervals over 5, 20 and 1 seed for Pendulum-v1, MountainCar-v0 and Hopper, respectively. In setups which include T, nan values prohibited the visualization of Pendulum-v1's performance early in training. For Pendulum-v1, no curriculum was applied since we did not find any curriculum to be sensible.

For long episodes, the recurrent forward pass of synthetic states through the transition function can lead to exploding values, which eventually overflow. This problem can be addressed by limiting the maximum episode length. Since most episodes are already extremely short in the T and TI setup (typically under 10 time steps) we set the maximum episode length to 1, effectively reducing the synthetic environment to an SCB task without transition dynamics, leading to the plain I setup. We find that this does not reduce the performance on any environment, with the exception of Pendulum-v1. However, the best performance of the 5 runs in TI and I is equal, and training can be stabilized by increasing the number of rollouts per population member.

A curriculum like in IC is needed to achieve competetive results on the brax environments. Similar curriculi can be introduced some to classic control environments. For example, decreasing the evaluation length from 1000 to 200 while meta-training an environment for MountainCar improves meta-training stability and performance.

Our setup includes two main hyperparameters: the latent distribution from which the initial states are generated and the curriculum. The second row of Fig. 3 shows meta-training curves for different latent distributions. We test four different latent distributions: a standard Gaussian, a uniform distribution over $[0, 1)$, a categorical uniform distribution, and a categorical distribution with probabilities $\text{softmax}([1, 2, \ldots, n])$, where $n$ is the dimensionality of the latent vector. When using categorical latent distributions, the initial state distribution becomes a categorical one as well and can be thought of as sampling from a set of meta-learned observations. Overall, the Gaussian and uniform distributions achieve a similar performance, outperforming the categorical ones. This is likely because they can densely sample a manifold of the state space. The third row of Fig. 3 shows meta-training curves for different curricula, showing that meta-training is robust to the choice of curriculum.

## 5 INTERPRETABILITY OF SYNTHETIC ENVIRONMENTS

The episodes in the synthetic environment can be limited to one step without a qualitative loss of performance (see Appendix A.1). In this case, the reward received is equal to the return, the state-,

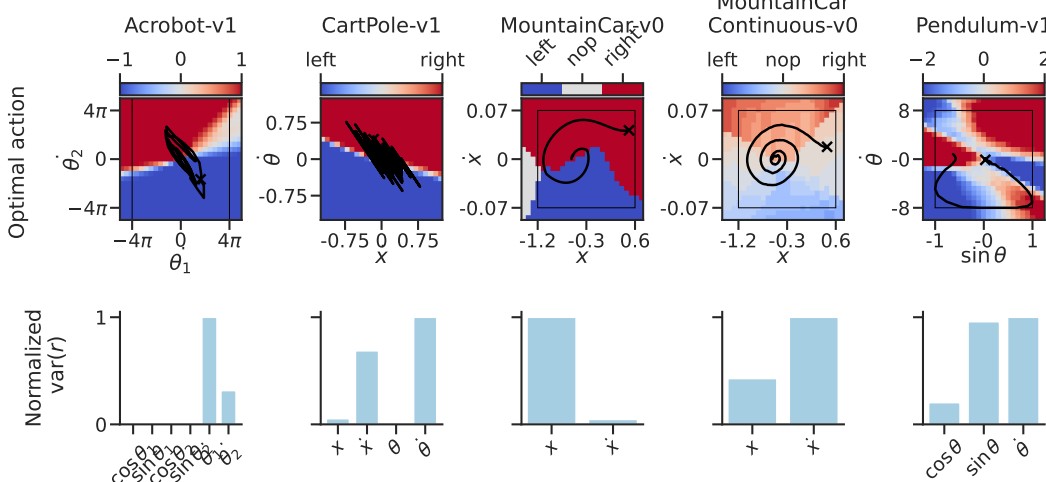

Figure 4: Synthetic environments provide interpretable insights into RL learning dynamics. **Top**. Optimal actions given the differentiable synthetic reward function for different states and 5 environments. We observe that the synthetic environment has discovered a type of state-action value function. Black box: observation space of the evaluation environment. Black line: representative trajectory in the real environment. Black x-marker: episode end. **Bottom**. Normalized variance in reward when varying part of the observation. Mean value over all observations in the space visualized in the top row.

and the state-action value function. This enables new ways to analyze the environment, such as easily finding the optimal action in each state via gradient descent or a simple grid search.

We visualize the optimal actions in the top row of Fig. 4. The resulting visualizations yield insights into the way that the synthetic environment trains an agent to perform a task: For example, the SCB for MountainCar-v0 never induces nops, since the return is highest if terminating early, while the optimal action in the MountainCarContinuous-v0 SCB is often close to nop since it includes a control cost instead of a constant negative reward.

Additionally, we can directly investigate the relationship between the observation and the return. We do so by fixing observation and action, and observing the variance in the reward when varying a single entry of the observation. The results are visualized in the bottom row of Fig. 4. We find that the reward is almost invariant to some parts of the observations. For example, varying the values of the angle in Acrobot-v1 has very little impact on the reward compared to the angular velocities. Similar findings hold for the position and angle CartPole-v1. Thereby we rediscover the results of Vischer et al. (2021); Lu et al. (2023a), who found the same invariances in the context of the lottery ticket hypothesis and adversarial attacks respectively, where these input channels were pruned or used to manipulate learning dynamics.

## 6    DOWNSTREAM APPLICATIONS POWERED BY SYNTHETIC ENVIRONMENTS

**Meta-learning with Synthetic Environments**. Our experiments demonstrate that synthetic environments are capable of training RL agents in faster wall clock time (see Fig. 1). But can they also be used to speed up downstream meta-learning? Here, we consider Learned Policy Optimization (LPO, Lu et al., 2022) and use a trained synthetic Pendulum environment to meta-learn a new RL objective function. In LPO, the parameters of a policy optimization objective are meta-evolved using the performance of trained agents. We find that the synthetic proxy is capable of training an objective that outperforms a PPO baseline *on the original environment* (see Fig. 5, left). In fact, the meta-training of LPO using the synthetic environment requires far fewer environment steps than training LPO using the real environment. Finally, the performance improvements do not only hold for environments used during meta-training, but also for the unseen Hopper environment.

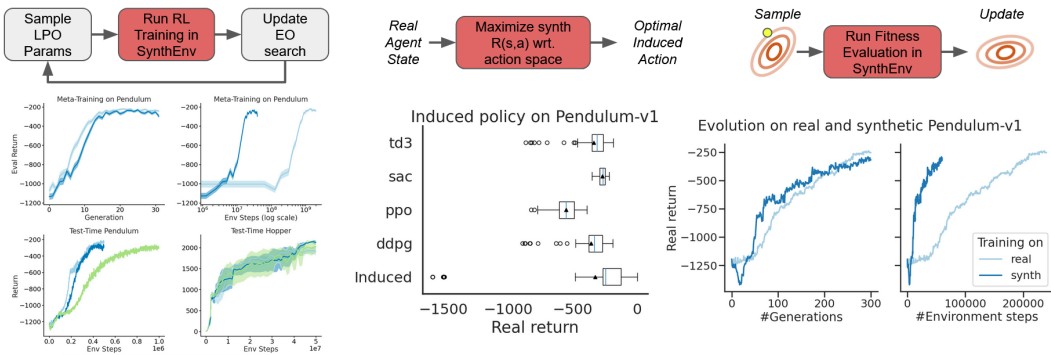

Figure 5: Downstream usability of synthetic environments. **Left**. Synthetic environments can be used for hardware-accelerated meta-learning, e.g. learned policy optimization (LPO, Lu et al., 2022) in which all meta-training is done in the synthetic environment. **Middle**. The discovered synthetic reward function can be directly used to extract an optimal policy, i.e. by computing the optimal action via $\arg\max_{a \in \mathcal{A}} R(s, a)$ from the one-step synthetic environment. Data of 100 episodes. **Right**. The discovered environment is capable of generalizing to non-gradient-based agent optimization using ES. IQM over 10 seeds.

**Extracting Optimal Policies from Synthetic Reward Functions**. A key advantage of our reward function parametrization is that it is differentiable with respect to the action space. Furthermore, given that the reward function was meta-optimized using single-step inner loop episodes, we find that it encodes a type of state-action value function. In fact, next we show that this can be utilized to decode an implicit optimal policy. More specifically, given an agent's state, we can compute an action choice by optimizing the reward function with respect to the action, $a^\star = \arg\max_{a \in \mathcal{A}} R_\theta(s, a)$. We call the resulting policy the 'induced' policy. In Fig. 5 (middle) we show that the resulting agent is capable of robustly solving the Pendulum task.

**Evolutionary Optimization with Synthetic Environments**. Finally, we investigated whether the SCB is tied to the specific RL algorithms it was meta-trained on. Instead, we find that it can be used in a very different optimization setting, using evolutionary black box optimization. In Fig. 5 (right) we find that a Pendulum MLP controller can be successfully trained using OpenAI-ES (Salimans et al., 2017) on an environment that was trained only with gradient based methods. Again, this demonstrates that the synthetic environment has not learned to 'hack' specific RL algorithms, but that it has captured general environment characteristics useful for training agents across paradigms.

## 7 CONCLUSION & DISCUSSION

**Summary**. We have demonstrated the successful discovery of SCBs capable of training RL agents that perform competitively in real environments. In order to do so we introduced various meta-optimization improvements, which enabled the successful meta-training. The SCBs yield insights into the relevance of individual observation entries and are easy to interpret. Furthermore, we showed that the SCB can be successfully deployed for various downstream applications including meta-learning, optimal policy derivation, and gradient-free agent optimization.

**Limitations**. While the meta-discovered environments are capable of generalizing across various training settings (e.g. type of algorithm and RL training hyperparameters), we find that the observed performance on the real environment can occasionally preemptively converge on more challenging tasks. This indicates a type of overfitting of the inner loop time horizon (Lange & Sprekeler, 2022). Hence, in these settings, the synthetic environment appears mostly suited for fast pre-training.

**Future Work**. Going forward we are interested in the discovery of synthetic simulators capable of promoting a truly open-ended learning process. Furthermore, we have focused on control environments with proprioceptive symbolic observation dimensions so far. A natural extension of our work is to pixel-based environments leveraging deconvolutional architectures for the initial state distribution.

## ETHICS STATEMENT

We find that neural networks are capable of representing various RL simulators in a compressed fashion. In principle, large models can therefore be capable of distilling data distributions and world models useful for self-training. Given that these systems are ultimately black-box, practitioners need to be careful when deploying them in real-world applications.

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

# Appendix

## Table of Contents

## A   ADDITIONAL BASELINES

### A.1   INFLUENCE OF THE EPISODE STEP LIMIT

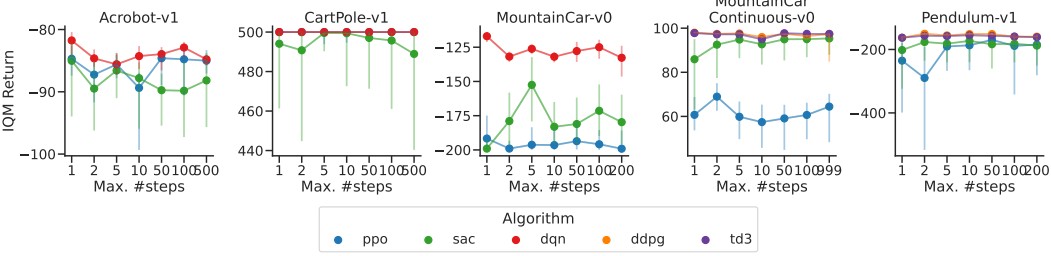

Figure 6: Return on classic control environments after training on synthetic environments with varying maximum episode lengths. We show the inner quantile mean with 95% confidence intervals of 20 agents, each evaluated on 50 seeds. The synthetic environments were meta-learned with sampled inner loop algorithms and hyperparameters. For exact values of the SCB (max. steps = 1) refer to Table 1.

| Algorithm | Acrobot-v1 | CartPole-v1 | MountainCar-v0 | MountainCar Continuous-v0 | Pendulum-v1 |
|---|---|---|---|---|---|
| PPO | -84.7 | 500.0 | -191.7 | 60.7 | -235.2 |
| SAC | -85.0 | 494.1 | -199.0 | 85.9 | -201.8 |
| DQN | -81.8 | 500.0 | -117.0 | – | – |
| DDPG | – | – | – | 98.0 | -163.0 |
| TD3 | – | – | – | 97.8 | -162.9 |

Table 1: Inner quantile mean of return on classic control environments after training 20 agents on an SCB (one step per episode). The SCB was meta-learned with sampled inner loop algorithms and hyperparameters.

### A.2   COMPARING SYNTHETIC REWARD AND $Q^*$

We show a comparison between optimal action in an SCB and optimal action according to an expert value learning algorithm in Fig. 7. While the experts achieve comparable performance, their value

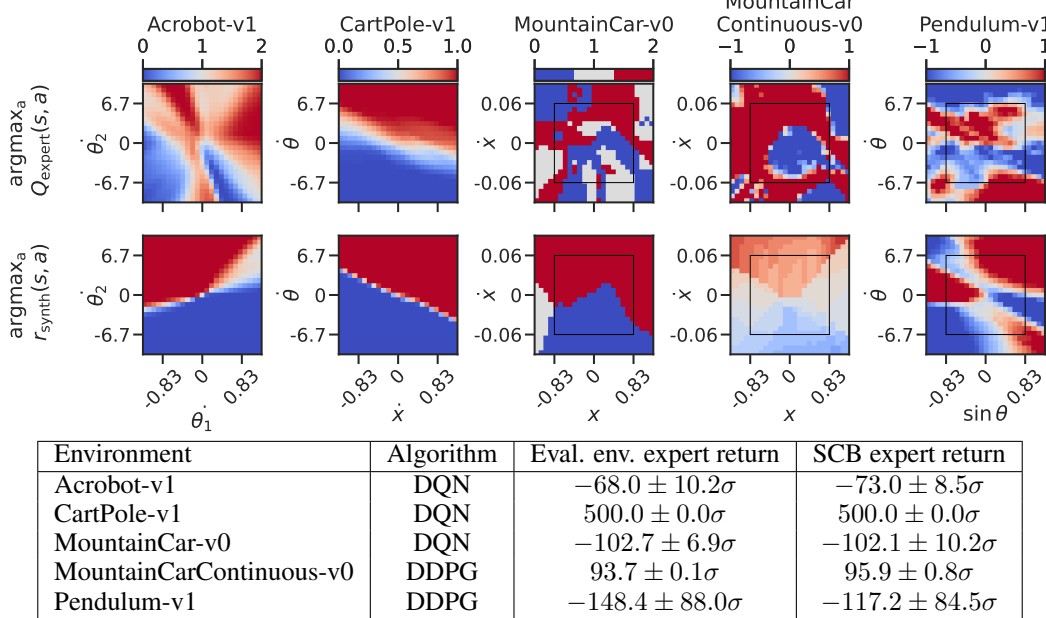

| Environment | Algorithm | Eval. env. expert return | SCB expert return |
|---|---|---|---|
| Acrobot-v1 | DQN | $-68.0 \pm 10.2\sigma$ | $-73.0 \pm 8.5\sigma$ |
| CartPole-v1 | DQN | $500.0 \pm 0.0\sigma$ | $500.0 \pm 0.0\sigma$ |
| MountainCar-v0 | DQN | $-102.7 \pm 6.9\sigma$ | $-102.1 \pm 10.2\sigma$ |
| MountainCarContinuous-v0 | DDPG | $93.7 \pm 0.1\sigma$ | $95.9 \pm 0.8\sigma$ |
| Pendulum-v1 | DDPG | $-148.4 \pm 88.0\sigma$ | $-117.2 \pm 84.5\sigma$ |

Figure 7: **Top**: Comparison between best action according to an evaluation evironment expert $\arg\max_a Q_{\mathrm{expert}}(s, a)$ (top row) and best action in the SCB $\arg\max_a r_{\mathrm{synth}}(s, a)$ (bottom row). The SCBs were trained with sampled hyperparameters and all algorithms in the inner loop. Experts were chosen to be value learning algorithms DQN and DDPG, respectively for discrete and continuous action environments. The experts were trained until convergence of their performance, and evaluated on 50 seeds. Training took 250.000 steps for DQN and 500.000 steps for DDPG. The parts of the observations on x and y axis were chosen based on their empirical importance (see Fig. 4). **Bottom**: Return of experts whose value function is visualized.

functions are not well interpretable in all cases. This is likely because the expert has a preferred set of trajectories from which he does not deviate often, such that it does not learn values of the states that are not visited (see Fig. 10). Additionally, the SCB can provide a more comprehensive support of the learned reward, since it can produce observations which are within the observation space but not within the manifold of possible observations. For example, Pendulum-v1 includes sine and cosine of an angle, which cannot simultaneously take a value of 1 in the evaluation environment, but can in the SCB.

In Fig. 8 we show that the synthetic reward differs from the value function of an expert in both in scale and state-dependence. To this end, we visualize $\max_a Q_{\mathrm{expert}}(s, a)$ of an expert policy and $\max_a r_{\mathrm{synth}}(s, a)$. A visual inspection of $\max_a Q_{\mathrm{expert}}(s, a)$ empirically confirms that it is close to the optimal state-value function. However, the values of the synthetic rewards are off by an order of magnitude, and it does not display the same relation to the state as $Q_{\mathrm{expert}}(s, a)$ does. While there seems to be some bias towards certain states, this bias might be arbitrary since the synthetic reward only has to consider the optimal action in order to induce optimal behavior.

We directly compare the value estimate of an expert and the synthetic reward along a trajectory in Fig. 9. The figure demonstrates that the meta-learned synthetic reward differs significantly from the value function of an expert policy. For example, the synthetic reward harshly punished nops in MountainCar-v0, since they don't lead to faster arrival at the goal. The expert value for the optimal action does not differ from non-optimal actions by much, since a nop does not lead to a state with much lower return. Another interesting observation can be made in MountainCarContinuous-v0, where the expert agent consistently chooses actions with a high absolute value, leading to a high control cost. This is punished by the synthetic reward, which shows a small increase whenever actions with a low control cost are chosen (clearly visible as spikes in the second row). The agent trained on the SCB takes much longer to reach the goal, but achieves an overall higher return by

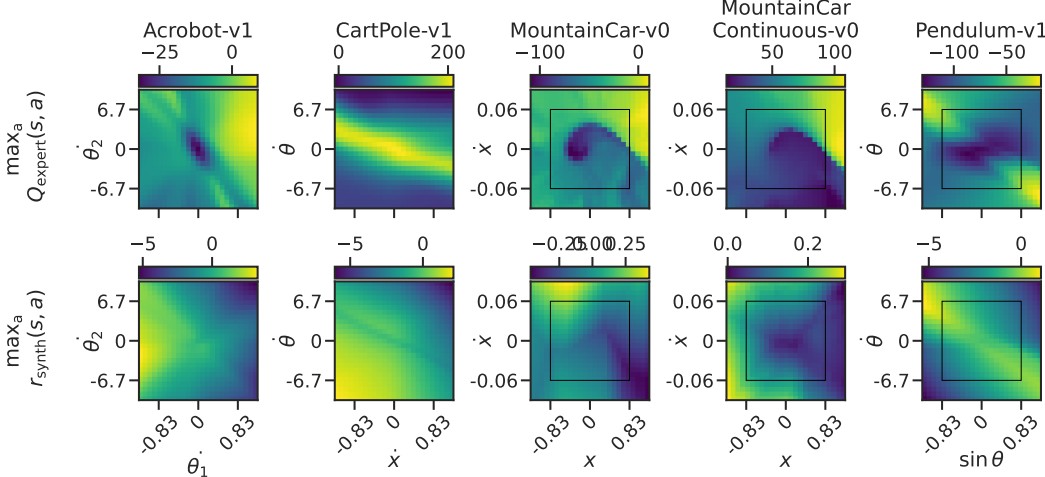

Figure 8: Comparison between $\max_a Q_{\text{expert}}(s, a)$ of an expert policy and $\max_a r_{\text{synth}}(s, a)$. For an expert with $Q \approx Q^*$, it holds approximately that $\max_a Q_{\text{expert}}(s, a) \approx V^*(s)$. The expert visualized is the same as in Fig. 7, which also shows final performance.

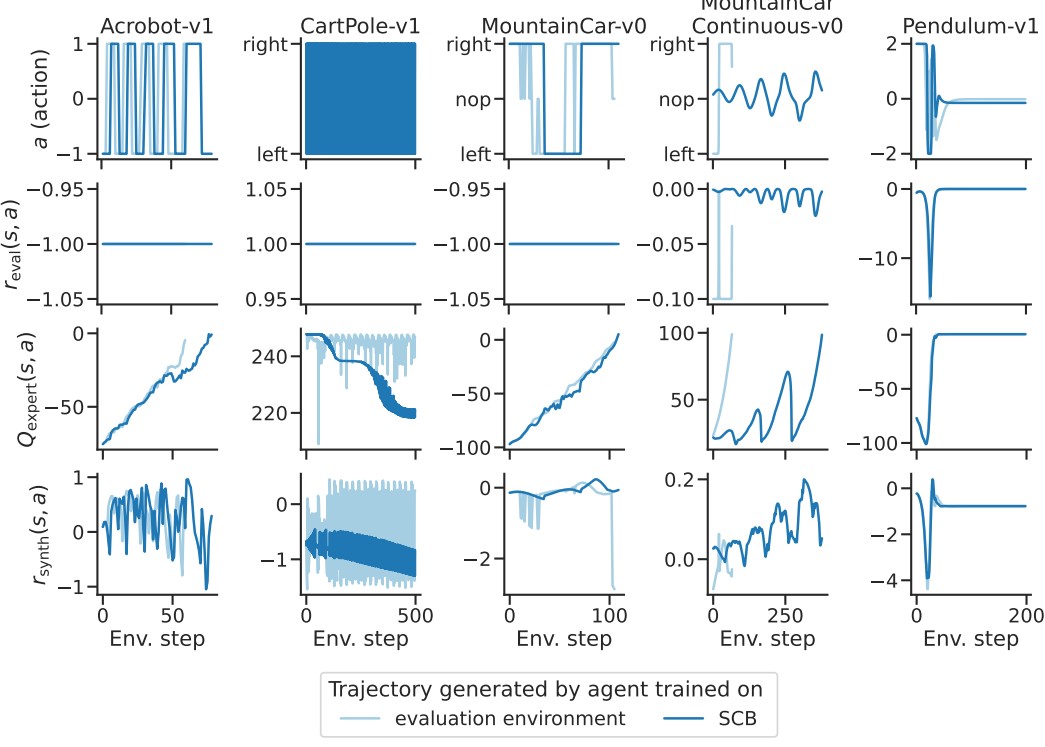

Figure 9: Comparison between $Q_{\text{expert}}(s, a)$ and $r_{\text{synth}}(s, a)$ along a trajectory of an agent trained on evaluation environment (expert) and SCB. The expert visualized is the same as in Fig. 7, which also shows final performance. The trajectories are generated on the same seed as the one visualized in Fig. 4. **First row**: Action taken in the evaluation environment. **Second row**: Reward obtained in the evaluation environment. **Third row**: State-action value estimate of a trained expert. **Fourth row**: Reward obtained in synthetic environment.

using less control force. These examples clearly show several degrees of freedom in the realization of the synthetic reward, enabling increased training speed.

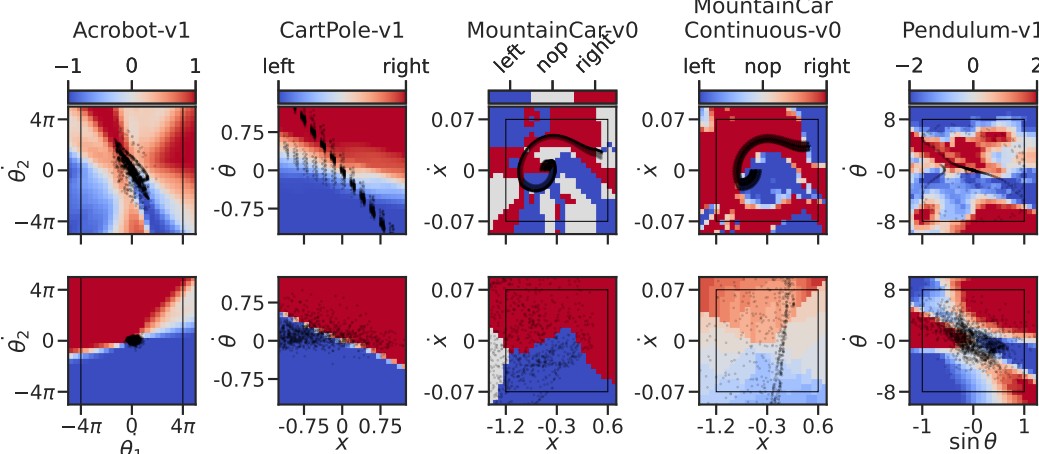

Figure 10: State distributions of an expert policy in the evaluation environment (top row) and the initial state in the SCB (bottom row). States of the expert were recorded for 2000 environment steps, automatically resetting the episode when terminated. 2000 states were randomly sampled from the initial state distribution of the SCB. The background color represents the best action according to the expert (top row) or best action in the SCB (bottom row), for details see Fig. 4. For details on the expert see Fig. 7.

We show the difference between the synthetic initial state distribution and the state distribution of an expert policy in Fig. 10. The expert policy only visits a small subset of possible states, while the distribution in the SCB is global and has a specific shape. We hypothesize that this distribution represents the states that are relevant to learning the task encoded by the evaluation environment, which can increase training speed and robustness. Additionally, the states sampled in the SCB might lie within the state space, but outside the manifold of possible states.

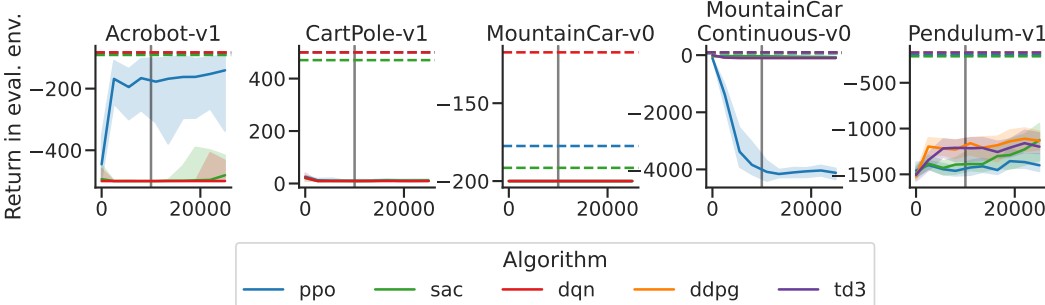

Figure 11: Training in an SCB where the synthetic reward has been replaced by the value function of an expert value learning agent. Dashed lines: IQM performance of an agent with sampled hyperparameters after training on and SCB. Vertical line: training cutoff in the inner loop. For details on the expert see Fig. 7.

Fig. 11 shows training curves for agents trained in and SCB where the synthetic reward has been replaced by the value function of an expert. In almost all cases, the agents achieve don't solve the environment. This is likely because the value function does not reward the best action significantly more than others, since there are no catastrophic rewards in the shown environments. As long as it the reward for the optimal action is the highest, the synthetic reward function can select arbitrary rewards for the other actions, which can increase training speed. Additionally, the synthetic reward was specifically optimized to generalize to all downstream algorithms, while the expert value function is only generated by a single expert and might not generalize to different algorithms.

**CartPole-v1**

| | | | | | | |
|---|---|---|---|---|---|---|
| PPO | × | | | × | × | × |
| SAC | | × | | × | | × |
| DQN | | | × | | × | × |
| PPO | 462.6 | 482.6 | 450.6 | 467.6 | 469.7 | 460.7 |
| SAC | 245.3 | 445.1 | 196.1 | 422.8 | 255.8 | 408.6 |
| DQN | 498.5 | 494.4 | 500.0 | 498.4 | 500.0 | 498.2 |

**Acrobot-v1**

| | | | | | | |
|---|---|---|---|---|---|---|
| PPO | × | | | × | × | × |
| SAC | | × | | × | | × |
| DQN | | | × | | × | × |
| PPO | -88.5 | -112.3 | -97.1 | -89.2 | -91.4 | -86.5 |
| SAC | -115.8 | -91.3 | -125.8 | -92.9 | -109.2 | -95.0 |
| DQN | -87.4 | -102.1 | -84.3 | -88.4 | -82.0 | -82.8 |

**MountainCar-v0**

| | | | | | | |
|---|---|---|---|---|---|---|
| PPO | × | | | × | × | × |
| SAC | | × | | × | | × |
| DQN | | | × | | × | × |
| PPO | -200.0 | -184.9 | -189.1 | -172.5 | -176.1 | -173.6 |
| SAC | -200.0 | -156.3 | -199.6 | -167.2 | -197.7 | -177.8 |
| DQN | -199.5 | -151.2 | -116.8 | -144.0 | -114.5 | -122.1 |

**MountainCarContinuous-v0**

| | | | | | | | | | |
|---|---|---|---|---|---|---|---|---|---|
| PPO | × | | | × | × | × | × | × | × |
| SAC | | × | | × | | × | × | × | × |
| DDPG | | | × | | × | | × | | × |
| TD3 | | | | × | | × | | × | × |
| PPO | 30.6 | -55.6 | -55.5 | -55.6 | 64.3 | 56.6 | 65.4 | 65.7 | 64.4 | 62.4 |
| SAC | 51.1 | -0.6 | -5.9 | -4.5 | 61.8 | 56.7 | 56.3 | 64.0 | 64.6 | 63.9 |
| DDPG | 67.2 | -0.0 | -2.1 | -1.0 | 93.3 | 89.3 | 96.5 | 96.1 | 92.4 | 97.2 |
| TD3 | 81.0 | -0.0 | -1.0 | -0.0 | 93.7 | 92.3 | 96.8 | 97.4 | 97.2 | 97.2 |

**Pendulum-v1**

| | | | | | | | | | |
|---|---|---|---|---|---|---|---|---|---|
| PPO | × | | | × | × | × | × | × | × |
| SAC | | × | | × | | × | × | × | × |
| DDPG | | | × | | × | | × | | × |
| TD3 | | | | × | | × | | × | × |
| PPO | -346.6 | -1397.5 | -1023.4 | -1178.0 | -309.8 | -378.2 | -304.0 | -328.4 | -331.1 | -299.9 |
| SAC | -490.8 | -304.5 | -861.8 | -482.1 | -345.0 | -492.7 | -432.4 | -359.4 | -382.2 | -355.9 |
| DDPG | -303.8 | -984.7 | -714.0 | -162.5 | -232.0 | -231.5 | -174.5 | -184.0 | -179.5 | -169.6 |
| TD3 | -259.6 | -1018.4 | -723.8 | -162.2 | -215.5 | -214.9 | -178.3 | -185.8 | -178.4 | -169.3 |

Table 2: Transfer of SCBs trained on a subset of RL algorithms. **Top half**: the × symbols in each column indicate if an algorithm was sampled while training the SCB. **Bottom half**: Final return after training an RL algorithm in the SCB, evaluated by training 100 agents and evaluating each on 50 seeds.

## A.3 GENERALIZATION ACROSS INNER LOOP ALGORITHMS

We show generalization and transfer across RL algorithms in Table 2. Overall, we note that not all algorithms need to be sampled in the inner loop to generalize. In the discrete action environments, using SAC alone leads to successful learning by all algorithms, while continuous action environments need only SAC and PPO. However, including all inner loop algorithms leads to a further performance increase, even beating the performance achieved by "overfitting" the SCB towards a single algorithm in almost all cases.

## B   HYPERPARAMETERS

We manually set the outer loop hyperparameters informed by exploratory experiments, since the experiments are very computationally expensive. The hyperparameters of the inner loop algorithms were set arbitrarily to reasonable values.

### B.1   OUTER LOOP HYPERPARAMETERS

|  | Classic Control | Pendulum | brax (*) |
|---|---|---|---|
| Init. SNES $\sigma$ | 0.05 | 0.05 | 0.05 |
| Population size | 128 | 64 | 256 |
| num. rollouts | 1 | 8 | 1 |
| num. eval. seeds | 50 | 50 | 16 |
| num. eval. seeds for population mean | 64 | 64 | 64 |
| multi algo. mode | all | all | sequential |

Table 3: Hyperparameters for meta-training. *multi algo. mode* refers to the way the RL algorithms are chosen in the inner loop. "All" means executing all available ones sequentially, and taking the mean of their returns as the fitness. "Sequential" means using algorithm $i$ where $i = \text{gen} \mod |A|$. (*) only applies to Hopper, Walker2D, Swimmer, Halfcheetah, Ant, the same parameters are used as in the classic control environments otherwise.

|  | num. generations |
|---|---|
| Acrobot | 300 |
| CartPole | 300 |
| MountainCar | 1000 |
| ContinuousMountainCar | 300 |
| Pendulum | 1000 |
| Inverted Pendulum | 300 |
| Inverted Double Pendulum | 300 |
| Reacher | 30 |
| Pusher | 100 |
| Hopper | 2000 |
| Walker2D | 2000 |
| Swimmer | 2000 |
| Halfcheetah | 2000 |
| Ant | 2000 |

Table 4: Number of generations for each environment

|  | MountainCar | brax (*) |
|---|---|---|
| type | linear | linear |
| init. eval. length | 1000 | 100 |
| final eval. length | 200 | 1000 |
| begin transition | 200 | 200 |
| num. transitions steps | 600 | 1600 |

Table 5: Hyperparameters for evaluation length curriculi. (*) only applies to Hopper, Walker2D, Swimmer, Halfcheetah, Ant, no curriculum is applied otherwise.

|  | all environments |
|---|---|
| network arch. | (32, ) MLP |
| activation | tanh |
| latent dist. | $\mathcal{N}(0, I_n)$ |
| latent size | (dim. of eval. envs. obs. space) |

Table 6: Hyperparameters for the synthetic environment

## B.2 INNER LOOP HYPERPARAMETERS

|  | Classic Control | brax |
|---|---|---|
| network arch. | (64, 64) MLP | (64, 64) MLP |
| activation | tanh (relu for Pendulum) | tanh |
| num. envs | 5 | 5 |
| num. steps | 100 | 100 |
| num. epochs | 10 | 10 |
| num. minibatches | 10 | 10 |
| time steps | $10^4$ | $10^4$ |
| max. grad. norm | 10 | 10 |
| learning rate | $\{0.01, \underline{0.005}, 0.001, 0.0005, 0.0001\}$ (without 0.01 for continuous environments) | 0.005 |
| discount | $\{1.0, \underline{0.99}, 0.95, 0.9, 0.8\}$ | 0.99 |
| $\lambda$ for GAE | $\{1.0, \underline{0.95}, 0.9, 0.8, 0.5\}$ | 0.95 |
| clipping $\epsilon$ | $\{0.1, \underline{0.2}, 0.3, 0.4, 0.5\}$ | 0.2 |
| entropy coef. | $\{0.0, \underline{0.01}, 0.05, 0.1, 0.5\}$ | 0.01 |
| value function coef. | $\{0.0, \underline{0.5}, 1.0, 1.5, 2.0\}$ | 0.5 |

Table 7: Hyperparameters for PPO in the inner loop. Underlined values are used in runs with a fixed configuration.

|  | Classic Control | brax |
|---|---|---|
| network arch. | (64, 64) MLP | (64, 64) MLP |
| activation | tanh (relu for Pendulum) | tanh |
| num. envs | 5 | 1 |
| buffer size | 2000 | 5000 |
| prefill buffer | 1000 | 1000 |
| batch size | 256 | 250 |
| grad. steps | 2 | 1 |
| time steps | $10^4$ | $10^4$ |
| learning rate | $\{0.01, \underline{0.005}, 0.001, 0.0005, 0.0001\}$ | 0.005 |
| discount | $\{1.0, \underline{0.99}, 0.95, 0.9, 0.8\}$ | 0.99 |
| Polyak $\tau$ | $\{0.99, \underline{0.95}, 0.9, 0.7, 0.8\}$ | 0.95 |
| target entropy ratio | $\{0.1, 0.3, 0.5, \underline{0.7}, 0.9\}$ | n.a. |

Table 8: Hyperparameters for SAC in the inner loop. Underlined values are used in runs with a fixed configuration.

| | Classic Control (discrete) |
|---|---|
| network arch. | (64, 64) MLP |
| activation | tanh |
| num. envs | 10 |
| buffer size | 2000 |
| prefill buffer | 1000 |
| batch size | 100 |
| grad. steps | 1 |
| time steps | $10^4$ |
| target update freq. | 50 |
| max. grad. norm | 10 |
| $\epsilon$ start | 1 |
| $\epsilon$ decay fraction | 0.5 |
| $\epsilon$ end | $\{0.01, 0.05, 0.1, 0.2\}$ |
| learning rate | $\{0.01, \underline{0.005}, 0.001, 0.0005, 0.0001\}$ |
| discount | $\{1.0, \underline{0.99}, 0.95, 0.9, 0.8\}$ |
| Double DQN $\tau$ | $\{\underline{\text{yes}}, \text{no}\}$ |

Table 9: Hyperparameters for DQN in the inner loop. Underlined values are used in runs with a fixed configuration.

| | Classic Control (continuous) | brax |
|---|---|---|
| network arch. | (64, 64) MLP | (64, 64) MLP |
| activation | tanh (relu for Pendulum) | tanh |
| num. envs | 1 | 1 |
| buffer size | 2000 | 5000 |
| prefill buffer | 1000 | 1000 |
| batch size | 100 | 100 |
| grad. steps | 1 | 1 |
| time steps | $10^4$ | $10^4$ |
| max. grad. norm | 10 | 10 |
| learning rate | $\{0.01, \underline{0.005}, 0.001, 0.0005, 0.0001\}$ | 0.005 |
| discount | $\{1.0, \underline{0.99}, 0.95, 0.9, 0.8\}$ | 0.99 |
| Polyak $\tau$ | $\{0.99, \underline{0.95}, 0.9, 0.7, 0.8\}$ | 0.95 |
| expl. noise | $\{0.1, \underline{0.2}, 0.3, 0.5, 0.7, 0.9\}$ | 0.2 |

Table 10: Hyperparameters for DDPG in the inner loop. Underlined values are used in runs with a fixed configuration.

|  | Classic Control (continuous) | brax |
|---|---|---|
| network arch. | (64, 64) MLP | (64, 64) MLP |
| activation | tanh (relu for Pendulum) | tanh |
| num. envs | 1 | 1 |
| buffer size | 2000 | 5000 |
| prefill buffer | 1000 | 1000 |
| batch size | 100 | 100 |
| grad. steps | 1 | 1 |
| time steps | $10^4$ | $10^4$ |
| max. grad. norm | 10 | 10 |
| learning rate | $\{0.01, \underline{0.005}, 0.001, 0.0005, 0.0001\}$ | 0.005 |
| discount | $\{1.0, \underline{0.99}, 0.95, 0.9, 0.8\}$ | 0.99 |
| Polyak $\tau$ | $\{0.99, \underline{0.95}, 0.9, 0.7, 0.8\}$ | 0.95 |
| expl. noise | $\{0.1, \underline{0.2}, 0.3, 0.5, 0.7, 0.9\}$ | 0.2 |
| target noise | $\{0.1, \underline{0.2}, 0.3, 0.5, 0.7, 0.9\}$ | 0.2 |
| target noise clip | $\{0.1, 0.4, \underline{0.5}, 0.7, 1.0, 1.3\}$ | 0.5 |

Table 11: Hyperparameters for TD3 in the inner loop. Underlined values are used in runs with a fixed configuration.

## C   SOFTWARE REQUIREMENTS & EXPERIMENT SIMULATION DETAILS

**Software**. All training loops and ES are implemented in JAX (Bradbury et al., 2018). All visualizations were done using Matplotlib (Hunter, 2007) and Seaborn (Waskom, 2021, BSD-3-Clause License). Finally, the numerical analysis was supported by NumPy (Harris et al., 2020, BSD-3-Clause License). Furthermore, we used the following libraries: Evosax: Lange (2022a), Gymnax: Lange (2022b), Brax: Freeman et al. (2021).

**Data Availability**. We provide access to two open source libraries in order to easily replicate our work:

- `synthetic-gymnax`: A repository of synthetic environments characterized by neural networks with pre-trained weight checkpoints.
- `purerl`: A set of hardware-accelerated RL algorithms (SAC, PPO, DQN, DDPG, TD3) that run entirely on GPU/TPU which enables fast meta-optimization evaluation.

**Compute Requirements & Experiment Organization**. The experiments were organized using the `MLE-Infrastructure` (Lange, 2021, MIT license) training management system.

Simulations were conducted on a high-performance cluster using between 1 and 20 independent runs (random seeds). We mainly rely on individual V100S and A100 NVIDIA GPUs.

The individual random search experiments last between 30 minutes and 48 hours, depending on the considered environment and choice of evolutionary optimization hyperparameters. The final evaluation of a tuned configuration, on the other hand, only requires a few minutes.

The tasks were chosen so that executing the entire benchmark only requires 2.5 days given 16 suitable GPUs. Furthermore, the open-data availability of the benchmark results allows researchers to focus on their method, instead of having to spend computing resources in order to collect baseline results.

