# OpenReview forum: "Discovering Minimal Reinforcement Learning Environments"
_ICLR.cc/2024/Conference — Submitted to ICLR 2024_

### Official Review · Reviewer_nrWg · 2023-10-31

**Soundness:** 2 fair
**Presentation:** 1 poor
**Contribution:** 2 fair
**Rating:** 3
**Confidence:** 4

**Summary:**

The work presents an extension to the synthetic environment learning framework. While keeping the meta-training loop the presented work proposes to learn initial state distributions instead of full transition dynamics when training synthetic environments. This approach is further extended with a curriculum learning approach to steadily increase evaluation rollout lengths in the target environment to stabilize the meta-training approach

**Strengths:**

The work presents an interesting extension to learning synthetic environments.
The idea to only learn an initial state distribution and fixing transition dynamics to always reach goal states is an interesting approach to environment design.
Further, the idea of using synthetic reward functions to extract an optimal policy that is learnable in the environment is very compelling as it provides a ground truth that is often not available. Thus, such learned environments could be used to provide more insights into the learning dynamics of different RL agents.

**Weaknesses:**

The work is often not clear enough in which parts are coming from the framework proposed by Ferreira et al. Thus, it took me some time to understand that basically the only change to the framework is the idea of learning an initial state distribution (with fixed dynamics) rather than learning the dynamics. This lack of clarity is particularly highlighted in the experiments where “the plain T setup” is declared as similar to Ferreira et al. but it is not elaborated on what the exact differences are. Further, in the experiments there is no clear comparison to the prior work and one is left to guess on the actual performance differences.

Figure 3 most often not show the learning curve for the proposed training setup. Only for hopper and Mountain car does it show the actual learning curve of the proposed IC training setup but for all other ablations only a dashed line of the final performance is shown. This omission of the learning curves seems rather strange and requires an explanation. Similarly, in this figure often the 95% confidence intervals seem wholly missing for some or even all methods. Lastly, in this ablation it seems rather strange to not include “no curriculum” as an option when showing the impact of different curricula. This is an important baseline to show. I understand that the difference in using curricula and no curricula is shown in the first row, but it is not easy to visually compare the ablated element in this way.

The writing could be improved as often statements are made which are not easy to follow. For example, already early in Section 3 it is stated that learning the transition dynamics is not as useful as only learning an initial state distribution but no reasons are given as to why. At least a reference to the later experiments would have shown that this statement is not just coming from thin air. Similarly, limiting an episode length to 1 seems to fall from thin air without any comparison if maybe a slight longer episode of 2 or 3 steps might perform better or not. It is simply stated that this does not lead to reduced performance except on pendulum, for which no explanation is given nor a hypothesis stated.
Another example is the statement that the SEs can train agents within 10k steps whereas in real environments this is often requires many more steps. Unfortunately, no learning curves are presented to visualize this difference. At least a reference to Fig 1 seems necessary to support this statement.

Overall the work seems interesting but not yet fully ready for publication.

**Questions:**

At the beginning of section 5 it is stated that “…, the episodes in the synthetic environment can be limited to one step without a quali- tative loss of performance. In this case, the reward received is equal to the return, the state-, and the state-action value function.“ Am I correct in understanding that learning an SE then is simply supervised learning of a reward function for particular policies? I don’t see why a learned initial state distribution is needed at all then. Would it not suffice to simply fix the initial distribution?

How did you set the hyperparameters of your method?

How much tuning is necessary to achieve good results?

---

> ### Author Response · Authors · 2023-11-19
>
> We thank reviewer nrWg for their suggestions on improving the paper, and will take every of their points into consideration. Please find our respective response below:
>
> >The work is often not clear enough in which parts are coming from the framework proposed by Ferreira et al. It is not elaborated on what the exact differences [between "the plain T setup" and Ferreira et al.] are.
>
> We have made the distinction clearer throughout the paper.
>
> In our experiments, its necessary to recreate Ferreira at al’s method within our framework, since we need to leverage our highly parallelizable implementation of RL algorithms to enable experiments on Brax. This means the “plain T setup” differs in the implementation of RL algorithms. We added clarification to the corresponding parts of the text.
>
> >Further, in the experiments there is no clear comparison to the prior work and one is left to guess on the actual performance differences.
>
> To our best knowledge, there is no work on meta-learning synthetic environments besides Fereirra et al., who only present results for CartPole and Acrobot. We benchmark against our implementation of their method in figure 3.
>
> >Figure 3 most often not show the learning curve for the proposed training setup.
>
> With the IC learning curve in each plot, they easily become cluttered. However, we acknowledge that this might make the plot less convincing, and created a new version including the curves with improved clarity. In Pendulum, we could not think of a sensible curriculum to apply, and therefore did not use one. We added a clarifying remark in the figure description.
>
> >[In figure 3] often the 95% confidence intervals seem wholly missing for some or even all methods.
>
> Because of the high computational cost, we only have one run per ablation in hopper, so no confidence intervals can be drawn.
>
> Note that while meta-evolution may only use a single seed, evaluation during meta-training uses 64 seeds.
>
> >Lastly, in this ablation it seems rather strange to not include “no curriculum” as an option when showing the impact of different curricula.
>
> We have included your suggestion about including “no curriculum” in the last row of figure 3.
>
> >The writing could be improved as often statements are made which are not easy to follow.
>
> We have improved the clarity of writing in a lot of places, thank you for your suggestions.
>
> >Limiting an episode length to 1 seems to fall from thin air without any comparison if maybe a slight longer episode of 2 or 3 steps might perform better or not.
>
> We included empirical results that compare the performance when using different episode lengths in the appendix.
>
>
> **Questions:**
> >Am I correct in understanding that learning an SE then is simply supervised learning of a reward function for particular policies? Would it not suffice to simply fix the initial distribution?
>
> We kindly ask the reviewer to clarify their question. During meta-training there is no direct access to the reward function or value function of an expert agent. Instead, the environment is optimized to maximize fitness. If they mean that the learned synthetic reward is similar to the optimal state-action value function, we are happy to point them towards our comprehensive comparison between both in the appendix of the revised version of our paper.
>
> Learning the initial distribution is critical to the success of the method. If you set the initial distribution to be that of the original environment, there are a lot of states which are never seen by the agent, and to which it cannot generalize. Parameterizing the initial distribution enables the CB to select observations which are relevant to learning the task encoded by the evaluation environment, and can therefore help with training speed.
>
> >How did you set the hyperparameters of your method?
>
> We manually set the outer loop hyperparameters informed by exploratory experiments, since the experiments are very computationally expensive. The hyperparameters of the inner loop algorithms were set arbitrarily to reasonable values, and remains very stable to their combinations. We included this information in the Hyperparameters section.
>
> >How much tuning is necessary to achieve good results?
>
> Our approach is relatively robust to outer loop hyperparameters. For example, we use the same hyperparameters for all Gym and small Brax environments. The exception is Pendulum, which needs a more stable fitness evaluation due to the very random nature of returns induced by favorable or unfavorable initial states. This leads to an increased number of rollouts per population member, and thereby a decreased population size due to memory constraints.
>
> We hope that we could answer your questions and improve our paper by addressing your concerns. Please reply if anything remains unclear.

---

> ### Author Response · Authors · 2023-11-21
> **Following up on the review of nrWg**
>
> We hope we have addressed the remarks of the reviewer and if so, that they would consider updating their score. We revised our paper such that
> - figure 3 includes all of the reviewers suggestions regarding its clarity
> - the difference between Ferreira et al. is made explicit in the relevant parts of the text
> - we include an empirical comparison of different episode lengths (figure 6)
> - we visualize the initial distribution of the SCBs in the appendix (figure 10)
> - statements are cleared up and include references to the relevant parts of the paper

---

### Official Review · Reviewer_rbXB · 2023-10-31

**Soundness:** 3 good
**Presentation:** 2 fair
**Contribution:** 3 good
**Rating:** 3
**Confidence:** 2

**Summary:**

This paper introduce a meta-optimization framework for synthetic environment discovery, where the parameters of the synthetic environment are meta optimized using SNES. Through extensive empirical study, the authors show that meta-training over a large range of inner loop tasks leads to synthetic environments that generalize across different RL algorithm and broad range of hyper-parameters. In addition, the authors shows that training contextual bandits achieves good enough result to generalize to the target environment. Compared to training as RL agents, contextual bandits are easy to interpret.

**Strengths:**

This paper present extensive experiment results to demonstrate and explain the proposed method. Firstly, the authors shows the performance of meta-trained synthetic environments on multiple openAI gym environments.  Further, the authors show that training contextual bandits is sufficient to train RL agents which significantly reduce the number of parameters needed through empirical study. The idea of simplify an RL problem to a CB by eliminating the transition probability makes the entire framework more feasible.  It does have a potential to be applied to more large scale real problems.

**Weaknesses:**

I found this paper very hard to follow. Some necessary background knowledge is not introduced in the paper.
Although this paper gives empirical proof for the statements, the intuition behind the ideas are now explained. Without any explanation, I am not convinced with the claimed results. For example, the authors claimed that the state transition probability is not necessary to train well-performing agents. I read the experimental result, but I cannot understand the intuition behind it.

**Questions:**

Similar to the points I made in weaknesses session, I would read more on the intuition behind all your experiments.

---

> ### Author Response · Authors · 2023-11-19
>
> We thank reviewer rbXB for their feedback and their recognition of our extensive empirical study.
>
> >Some necessary background knowledge is not introduced in the paper.
>
> As the reviewer summarized, we discover synthetic environments (contextual bandits) using a meta-learning with an evolutionary strategy. We do so by taking advantage of recent advances in hardware accelerated RL.
> Section 2 already contains background on 1- reinforcement learning (RL), 2- curricula in RL, 3- RL from synthetic data, 4- meta-optimization, 5- evolutionary strategies, including evolutionary meta-learning and 6- hardware-accelerated RL. We argue this covers all required background for our paper. If the reviewer believes we omitted something, we ask that they please let us know what specifically, so that we may add it to the revised manuscript.
>
> >Although this paper gives empirical proof for the statements, the intuition behind the ideas are now explained.
>
> We would appreciate if the reviewer could provide additional examples on which ideas they find lack intuitions. They do provide one example, which we address below, but they seem to imply there are more.
>
> >[...] Without any explanation, I am not convinced with the claimed results.
>
> We are not sure what the reviewer means here. Is the claim that because our intuitions are not sufficiently explained, the experiments are invalidated?
>
> >The authors claimed that the state transition probability is not necessary to train well-performing agents. I read the experimental result, but I cannot understand the intuition behind it.
>
> We have updated the manuscript to make the connection between single step MDPs and contextual bandits more explicit, and give an intuition on why the optimal policy of any MDP can be found by training on a CB. We have also clarified various sections throughout the paper.
> We additionally added a more comprehensive discussion of our choice to use contextual bandits to the method section, see "Synthetic Environment Setup".
>
> If the reviewer still finds that some sections are hard to follow, we ask that they please specify which, so that we may improve the paper further.

---

> ### Author Response · Authors · 2023-11-21
> **Following up on the review of rbXB**
>
> We hope we could address the reviewer's questions by updating our manuscript, which now includes a dedicated part of the method section to justify our usage of contextual bandits, including the intuition why training on CBs leads to successful transfer to MDPs. Additionally, we added several insightful visualizations of SCB components in the appendix, such as the initial state distribution (figure 10), and the synthetic reward function as compared to the value function of an expert agent (figures 7-9).
>
> If any questions remain, we kindly ask the reviewer to continue the discussion so that we may address them. If this is not the case, we politely request for them to update their score to reflect our improvements.

---

### Official Review · Reviewer_XskH · 2023-11-01

**Soundness:** 2 fair
**Presentation:** 3 good
**Contribution:** 2 fair
**Rating:** 3
**Confidence:** 4

**Summary:**

This paper proposes a meta-learning approach to learning a synthetic model of the RL environment as a proxy for finding the optimal policy, instead of directly interacting with the real environment. The authors formulate the problem as a meta-learning problem and solve it using evolution strategy (ES). To overcome the difficulty of learning the dynamics of the actual environment, the authors propose that learning a synthetic *contextual bandit* (SCB) model is sufficient to achieve the goal and train the agent policy. They conduct experiments and provide ablation studies to analyze the choice and variants of the proposed method.

**Strengths:**

1. Using a simpler model (synthetic contextual bandit, SCB) as a proxy is an interesting idea when environment is complex.
2. Experiments on interpretability justify the choice of SCB as the proxy.

**Weaknesses:**

One of the main contributions of this paper is the use of a synthetic contextual bandit (SCB) as a proxy to the real environment. However, it is important to note that:

* A contextual bandit (CB) can be converted to a Markov decision process (MDP), but not vice versa, because CB is stateless. This means that the SCB model may not be able to accurately capture the dynamics of more complex environments, such as Go, where state is essential for planning and decision-making.
* It is also unclear how synthetic contextual bandit can work for partial observation environments, where the agent does not have access to all of the state information.

In other words, the SCB model may be able to learn to play simple games, such as Atari Breakout, where the state space is relatively small and statelessness is not a major issue. However, for more complex games, such as Go, where the state space is very large and statelessness is a major issue, the SCB model is unlikely to be able to learn to play at a high level. Additionally, it is unclear how the SCB model would perform in partial observation environments.

In addition to the limitations of the SCB model discussed above, there are several other issues with this paper:

* One important paper is not cited nor discussed, which is closely related:

Ferreira et al. (2022) proposed a similar approach of learning synthetic environments and reward networks for reinforcement learning. It would be helpful to discuss the relationship between this work and the proposed method.

* There is no comparison with the state-of-the-art results on the environments/tasks in this paper. It is understandable if the proposed method does not outperform the state of the art, as the inner policy optimization can be different. However, it would be interesting to see how the proposed method compares to other methods on the same tasks.

* The motivation for using a synthetic environment is not entirely clear. In the first paragraph of the introduction, the authors mention the biological fact that organisms can learn from artificial stimuli. However, it is not clear how this relates to the use of synthetic environments in reinforcement learning. Additionally, the authors do not explain why learning or meta-learning RL policies is not sufficient.

* The colors in Figure 5 should be revised so that synthetic curves have the same color, and the same for the real curves. This would make the figure easier to read and interpret.

Overall, the use of a SCB as a proxy to the real environment is a promising approach, but it is important to be aware of its limitations. For more complex environments, such as Go, and partial observation environments, models like MCTS are still necessary for learning to play at a high level. The authors need to address the issues raised above in order to strengthen their work.

Reference

Ferreira, F., Nierhoff, T., Sälinger, A., & Hutter, F. *Learning Synthetic Environments and Reward Networks for Reinforcement Learning*. In ICLR 2022.

**Questions:**

1. Why do the authors choose to evaluate the proposed method on Brax environments, instead of Gym or other popular environments?

2. The authors claim that limiting the episode length to one step in the synthetic environment does not qualitatively affect the performance of the agent (Section 4, last subsection; Section 5, beginning). However, I could not find any figure or table in the paper that shows this result. Can the authors please provide this information?

3. The authors claim that "even state-of-the-art RL algorithms such as PPO struggle with solving MountainCar" (Section 1, second paragraph). However, I am not sure if this is true. Can the authors please provide citations to support this claim?

---

> ### Author Response · Authors · 2023-11-19
>
> We thank reviewer XskH for their comments and suggestions, and invite them to read our response to their points below.
>
> > A contextual bandit (CB) can be converted to a Markov decision process (MDP), but not vice versa, because CB is stateless. This means that the SCB model may not be able to accurately capture the dynamics of more complex environments, such as Go, where state is essential for planning and decision-making. It is also unclear how synthetic contextual bandit can work for partial observation environments, where the agent does not have access to all of the state information.
>
> While environments with temporal dynamics are better modeled by MDPs, training a agent on a CB can imprint the policy of the optimal agent on the MDP. This can be done by setting the reward function to the value function of the optimal agent, and sampling the observation space sufficiently. By maximizing the reward in the CB, the agent will therefore maximize the value in the real environment. We recognize that this point was not addressed clearly enough in the paper, but the method remains general to MDPs, including Go. We added an explanation of this relation in our methods section (see "synthetic environment setup").
>
> Partial observations can be handled in the same way, since the CB can present partial observations.
>
> >One important paper is not cited nor discussed, which is closely related: Ferreira et al. (2022)
>
> We mistakenly referenced an earlier version of Ferreira et al. (2022), and have now changed it in favor of the version published in ICLR. We benchmark against their method in figure 3 and show clear improvements resulting from our contributions.
>
> >There is no comparison with the state-of-the-art results on the environments/tasks in this paper.
>
> We are using common SotA RL algorithms in the inner loop, and compare to the current SotA work on meta-learning synthetic environments by Ferreira et al. For a comparison to training directly on the evaluation environment please refer to figure 1.
>
> >The motivation for using a synthetic environment is not entirely clear. In the first paragraph of the introduction, the authors mention the biological fact that organisms can learn from artificial stimuli. However, it is not clear how this relates to the use of synthetic environments in reinforcement learning.
>
> In the biological analogy, artificial stimuli correspond to the synthetic environment, while the real world corresponds to the evaluation environment. Before birth, the agents are pretrained in a way that transfers well to the real world, similar to our synthetic training, which transfers well to the evaluation environment.
>
> Besides this biological analogy, we list several motivations for training a synthetic environment, including computationally efficient simulation, faster RL training, no need for hyperparameter tuning, and the application to several downstream tasks.
>
> >Additionally, the authors do not explain why learning or meta-learning RL policies is not sufficient.
>
> Learning RL policies on an evaluation environment directly is sufficient for achieving a well-performing agent. However, training synthetic environments has several additional benefits, such as enabling the downstream tasks we present in section 6.
>
> >The colors in Figure 5 should be revised so that synthetic curves have the same color, and the same for the real curves. This would make the figure easier to read and interpret.
>
> We have changed the colors of figure 5 to be consistent.
>
> We hope to have sufficiently addressed the reviewers points. If anything remains unclear, we are happy to continue the conversation.

---

> ### Author Response · Authors · 2023-11-21
> **Following up on the review of XskH**
>
> We hope our comments have addressed the reviewer's concerns and if so, that they would consider updating the score. Also note that we have updated a our manuscript by
> - including their suggestions on figures and references, and clarifying statements across the whole paper
> - dedicating a part of the method section specifically to show that training on an SCB can lead to the optimal policy of an MDP
> - adding a comprehensive comparison between meta-learned reward and optimal value function on the evaluation environment

---

### Official Review · Reviewer_ABpb · 2023-11-01

**Soundness:** 2 fair
**Presentation:** 3 good
**Contribution:** 2 fair
**Rating:** 3
**Confidence:** 4

**Summary:**

This paper proposes to parameterize and optimize RL environments, thus "discovering" synthetic environments.
It is also remarked that this problem can be made much simpler by not learning a transition function for the synthetic environment, limiting the synthetic environment to a contextual bandit while preserving performance benefits.
Empirical results show that the synthetic environment can be used to train a variety of learning algortihms by meta-learning the synthetic environment over a distribution of inner-loop algorithms (different algorithms, as well as different hyperparameters).
The synthetic environments "are interpretable" and have other applications in down-stream tasks.

# Decision
While I like the overall direction of this research, I think there are several problems with the specifics of its execution and must recommend rejection.
The major flaw in this paper is that it seems to solve for the value function in a much more complicated way using meta-learning, without clearly demonstrating the benefits of doing so.
Some claims are not substantiated in the paper at all, and several other contributions are not clearly conveyed.
The paper would require a more careful treatment of the relevant baselines, as well as careful motivation for the proposed method, along with more evidence to support the claims.

**Strengths:**

- The research direction considered, discovering environments in which algorithms can learn quickly, seems novel and has potential to be quite interesting for deployment of RL algorithms. I especially like that use on further downstream tasks, beyond the prescribed inner-loop, are considered. This can have several benefits not only for training RL algorithms but in developing new algorithms with discovered environments for quicker iteration.

**Weaknesses:**

- There seems to a connection between the proposed method and simpler non-meta RL algorithms that just learn the value function, which is only mentioned in passing. The optimal reward of the contextual bandit seems to be the optimal value function at the context (state in the MDP). What is the benefit of meta-learning this, rather than just using monte-carlo returns? Or, what is the benefit of this over just learning the value function itself?
- Important claims made at the beginning of the paper, such as improved wall-clock time, are not substantiated.
- Overall, the empirical results fail to convince me of the importance of the contribution. The results in Figure 2 show that training in the synthetic environment is feasible to learn a policy (which is interesting, but also not entirely surprising because of the first point). The results, however, do not demonstrate that this provides a benefit over training in the real environment. The bottom row of figure 2 shows that baseline algorithms are underperformant on some hyperparameter distribution. But the performance on a particular hyperparameter distribution is also not entirely relevant. The benefit of an RL algorithm that performs well on a distribution of hyperparameters is robustness, and results in a less computationally expensive hyperparameter sweep. But the cost of this sweep can be much lower than the cost of the hyperparmaeter sweep and subsequent meta-learning of the synthetic environment itself. Your setup involves many hyperparameter choices in both the inner and outer loop. These design decisions are not abalated and do not provide enough confidence that the empirical results are reliable.

**Questions:**

- Section 2 (Curricula): you state that you learn a curriculum in this section, but in contribution 1 it seems that you in-fact design a curriculum. These two claims seem contradictory.
- Section 2 (Synthetic Data): I understand that there is some prior work that attempts to train reinforcement learning algorithms from synthetic data, but I do not think this is well-motivated. Why not just use the real data? You need it for evaluation anyway to learn the synthetic enviornment, so you may as well use it during meta-learning as well.
- In your experiments, is the meta-learned synthetic environment tuned to a specific real environment?
- Results, figure 3: why is there a discrepancy between the evaluation of the ablation and the results in Figure 2. Specifically for hopper? I understand ablations can be expensive, and may necessitate a smaller set of results, but this ablation does not explain the performance differences with respect to the original results.
- I do not see how synthetic environments provide any interpretability beyond what a learned value function can provide. It would be interesting to see whether the reward function meta-learned for synthetic environments differs in some qualitative way from that of a learned value function. But otherwise, this analysis does provide me with any additional insight.

---

> ### Author Response · Authors · 2023-11-19
>
> We thank reviewer ABpb for their insightful comments and are happy to improve the paper by addressing their points.
>
> >The optimal reward of the contextual bandit seems to be the optimal value function at the context [...]. What is the benefit of meta-learning this?
>
> When setting the synthetic reward (SR) to be the optimal value function of the target environment, agents trained in the SCB are guaranteed to transfer.
> However, the SR does not have to correspond to the optimal value function to achieve a high performance. RL agents that learn the value typically aim to choose actions which maximize it. This means that if $argmax_a Q(s,a)$ is equal between SCB and target environment, the behavior of trained agents should be the same.
> Thus the SR can be interpreted as a shaped reward. This has several advantages such as increased training speed, since the SR has an influence on the parameter updates.
> We acknowledge that this point has not been sufficiently addressed in the paper, and have added a dedicated part of section 3 (see "Synthetic Environment Setup"), as well as an empirical comparison to the value function in the appendix.
>
> > Important claims made at the beginning of the paper, such as improved wall-clock time, are not substantiated.
>
> We show a significant increase in simulation speed and an example of wall clock training curves for halfcheetah in figure 1. We added more information on these experiments to the caption.
>
> >The results, do not demonstrate that [training in the SCB] provides a benefit over training in the real environment.
>
> The significant increase in simulation speed can be leveraged in several ways, e.g. in neural architecture search. Even if training on the evaluation environment instead of the SCB achieves a higher reward, the SCB can be used for rapid pretraining. The benefits of synthetic environments in potential downstream applications are demonstrated in section 6.
>
> >The cost of [a] sweep can be much lower than the cost of the hyperparmaeter sweep and subsequent meta-learning of the synthetic environment itself.
>
> You are right that running a hyperparameter sweep is less cost efficient than meta-learning the synthetic environment. However, it has not been our goal to train a single well-performing agent, but to train the synthetic environment itself. This holds value since it can be used in a myriad of downstream tasks, for example those shown in section 6.
>
> >Your setup involves many hyperparameter choices in both the inner and outer loop. These design decisions are not abalated.
>
> The inner loop hyperparameters are sampled from a large range of sensible values. The performance on Brax tasks with fixed hyperparameters is shown in figure 1. Including parameters outside of this range would often lead to numerical instabilities, which is why we chose not to include it as an ablation.
> Since our experiments are very computationally costly, we aimed to ablate the most relevant and newly contributed outer loop parameters in figure 3. We would kindly ask the reviewer to clarify what they mean by design decisions. What other parameters are of interest to them?
>
>
> **Questions:**
> > Section 2 (Curricula): you state that you learn a curriculum in this section, but in contribution 1 it seems that you in-fact design a curriculum.
>
> In section 2 we state that "we use meta-optimization to discover state reset and state-action-dependent reward functions maximizing downstream learning progress". The reset and reward functions are not inner loop curricula. The curriculum mentioned in our contributions refers to a curriculum on the outer loop. We changed the paragraph in the background section such that this becomes clear.
>
> >Section 2 (Synthetic Data): Why not just use the real data [instead of synthetic data]?
>
> In many cases synthetic data is used because it is cheaper to generate. After meta-training, this is also the case for our SCBs (see figure 1). We added this motivating example to the relevant paragraph in section 2.
>
> >In your experiments, is the meta-learned synthetic environment tuned to a specific real environment?
>
> We are unsure what the reviewer means by this question. In our experiments, we train a separate synthetic environment per target environment.
>
> >Results, figure 3: why is there a discrepancy between the evaluation of the ablation and the results in Figure 2. Specifically for hopper?
>
> Figure 2 shows the results of meta-training with sampled inner loop algorithms, while Figure 3 shows a fixed inner loop algorithm (see column title). We added this information to the caption to clear up any confusion.
>
> >I do not see how synthetic environments provide any interpretability beyond what a learned value function can provide.
>
> When given a value function with the same parameterization as the synthetic reward, it can be analyzed in the same way. However, the two functions differ (see above).
>
> We hope to have answered the reviewers questions and are happy to engage in further discussion.

---

> ### Comment · Reviewer_ABpb · 2023-11-21
> **Thanks for the reply!**
>
> Thank you for the reply, I see this paper more favourably given your responses but I still think it is below the acceptance threshold. Here is some of my feedback regarding individual comments:
>
> > Meta-learned synthetic reward as learned shaping
>
> This is an interesting perspective, and one that I think has merit. But it raises the question: what is the benefit of meta-learning the shaped reward in this way vs some other meta-learning approach to reward shaping?
>
> > improved wall-clock time
>
> It is good that training in the SCB lowers the number of env steps needed to learn a performany policy. It also makes sense that the cost of running the SCB is lower because no state transition is needed. But, I think the paper should demonstrate the meta-learning the SC + learning in the environment is comparable to baselines. Otherwise, an argument has to be made that the meta-learned SCB provides other value (beyond, say, an optimal value function).
>
> > In our experiments, we train a separate synthetic environment per target environment.
>
> This is what I was wondering, because one of the benefits of meta-learning is generality. It needs to be demonstrated that meta-learning the SCB + learning in the SCB provides some benefit over learning in the original env alone.

---

> > ### Author Response · Authors · 2023-11-21
> >
> > We thank the reviewer for continuing the discussion and hope to address their remaining concerns.
> >
> >
> > > What is the benefit of meta-learning the shaped reward in this way vs some other meta-learning approach to reward shaping?
> >
> > Other approaches such as Ref. [1] or Ref. [2] assume a designed reward shaping function and learn its parameterization.
> > The method of reward networks from Ferreira et al. [3] and our method both use a neural network to parameterize the reward, opening a much larger space of possible learned reward functions. A comparison to reward shaping via an Intrinsic Curiosity Module (see Ref. [4]) and count-based exploration (see Ref. [5]) can also be found in Ref. [3].
> >
> > > The paper should demonstrate the meta-learning the SC + learning in the environment is comparable to baselines. Otherwise, an argument has to be made that the meta-learned SCB provides other value (beyond, say, an optimal value function).
> >
> > We achieve comparable performance on classic control environments, and even beat some scores achieved by experts trained on the evaluation environment directly (see figure 7).
> > The performance of SCB-trained agents on Brax is often lower than the evaluation environment expert. However, its still far above a random baseline, and shows clear training success.
> >
> > Additionally, we argue that the downstream applications in section 6 provide more than enough value to justify training of SCBs. We successfully demonstrated the usage in a meta-learning application (LPO), where an RL algorithm that was meta-learned on the SCB outperformed the PPO baseline when training an agent _on the original environment_. Additionally, we show that the SCB has generalized to neuroevolution, an entirely new learning algorithm, enabling the same computational speedup on a previously unseen task. Other possible applications include rapid architecture search or pretraining. The one-time cost of training an SCB can therefore be recouped.
> >
> > These applications would not be possible with an optimal value function alone, since the initial state distribution is necessary to train RL agents. Additionally, training RL agents with only the value function instead of a meta-learned reward is not sufficient (see figure 11).
> >
> >
> > >One of the benefits of meta-learning is generality. It needs to be demonstrated that meta-learning the SCB + learning in the SCB provides some benefit over learning in the original env alone.
> >
> > While we fix the synthetic environment to one evaluation environment, it is general over inner loop algorithms and hyperparameters (which is our task distribution). We show generalization from gradient-based RL to neuroevolution, an entirely different optimization scheme, in section 6. This degree of generalization is not present when only training on a single hyperparameter configuration (see figure 2) or a single algorithm (see table 2 in the appendix of our new revision, also the limited degree of transfer from DDQN to TD3 reported in Ref. [3]).
> >
> > It is not our main goal to beat the performance of an expert agent on an evaluation environment, but we are interested in the SCB and its applications itself. As stated above, we believe that the many possible applications for the SCBs hold more than enough value to justify the one-time investment of meta-training. To this end we are publishing the SCB checkpoints, such that anyone can use them in their project.
> >
> > We hope to have addressed the reviewer's concerns, and cleared up any confusion regarding the relevance of our contributions.
> >
> > **References**
> > - [1] Y. Hu, W. Wang, H. Jia, Y. Wang, Y. Chen, J. Hao, F. Wu, and C. Fan. Learning to utilize shaping rewards: A new approach of reward shaping. In Proc. of NeurIPS’20 (2020)
> > - [2] A. Faust, A. Francis, and D. Mehta. Evolving rewards to automate reinforcement learning. In ICML Workshop on Automated Machine Learning (2019)
> > - [3] F. Ferreira, T. Nierhoff, A. Sälinger, and F. Hutter. Learning Synthetic Environments and Reward Networks for Reinforcement Learning. In International Conference on Learning Representations (2022)
> > - [4] D. Pathak, P. Agrawal, A. A. Efros, and T. Darrell. Curiosity-driven exploration by self-supervised prediction. In Proc. of CVPR’17 (2017)
> > - [5] A. L. Strehl and M. L. Littman. An analysis of model-based interval estimation for markov decision processes. Journal of Computer and System Sciences (2008). ISSN 0022-0000

---

### Meta-Review · Area_Chair_Jr6e · 2023-12-14

**Metareview:**

The underlying idea of this paper - automatically generating environments that help reinforcement learning agents generalize. The surprising part is how simple the environments are, they are just contextual bandits. As one reviewer points out, this should limit the applicability of the technique to essentially Markovian problems. That the method works at all is certainly impressive.

It is is instructive that the reviewers universally argue that the paper should not be accepted, but that they mostly complain about different thing. To me, that is a sure sign that the paper is not getting across what it wants to get across. And indeed, I found it very hard to follow. I think that you can take the comments from the reviewers into account and when you rewrite the paper for another venue focus on clarity.

A slight comment on the related work is there is quite a bit of work within the games community on Procedural Content Generation that would be strongly related to what you do in this paper, perhaps Bontrager's Generative Playing Networks is the closest.

**Justification For Why Not Higher Score:**

Too dense (me and/or the paper).

**Justification For Why Not Lower Score:**

N/A

---

### Decision · Program_Chairs · 2024-01-16

Reject